# Intelligent Classification of Japonica Rice Growth Duration (GD) Based on CapsNets

**DOI:** 10.3390/plants11121573

**Published:** 2022-06-15

**Authors:** Xin Zhao, Jianpei Zhang, Jing Yang, Bo Ma, Rui Liu, Jifang Hu

**Affiliations:** 1College of Computer Science and Technology, Harbin Engineering University, Harbin 150086, China; zxxsnh@hrbeu.edu.cn (X.Z.); yangjing@hrbeu.edu.cn (J.Y.); 2College of Computer and Control Engineering, Qiqihar University, Qiqihar 161006, China; 3Qiqihar Branch of Heilongjiang Academy of Agricultural Sciences, Qiqihar 161006, China; mabo8210@haas.cn (B.M.); hujifang7@haas.cn (J.H.); 4College of Agricultural Engineering, Heilongjiang Bayi Agricultural University, Daqing 163000, China; liurui@byau.edu.cn

**Keywords:** japonica rice, Raman spectroscopy, Python, capsule networks, growth duration

## Abstract

Rice cultivation in cold regions of China is mainly distributed in Heilongjiang Province, where the growing season of rice is susceptible to low temperature and cold damage. Choosing and planting rice varieties with suitable GD according to the accumulated temperate zone is an important measure to prevent low temperature and cold damage. However, the traditional identification method of rice GD requires lots of field investigations, which are time consuming and susceptible to environmental interference. Therefore, an efficient, accurate, and intelligent identification method is urgently needed. In response to this problem, we took seven rice varieties suitable for three accumulated temperature zones in Heilongjiang Province as the research objects, and we carried out research on the identification of japonica rice GD based on Raman spectroscopy and capsule neural networks (CapsNets). The data preprocessing stage used a variety of methods (signal.filtfilt, difference, segmentation, and superposition) to process Raman spectral data to complete the fusion of local features and global features and data dimension transformation. A CapsNets containing three neuron layers (one convolutional layer and two capsule layers) and a dynamic routing protocol was constructed and implemented in Python. After training 160 epochs on the CapsNets, the model achieved 89% and 93% accuracy on the training and test datasets, respectively. The results showed that Raman spectroscopy combined with CapsNets can provide an efficient and accurate intelligent identification method for the classification and identification of rice GD in Heilongjiang Province.

## 1. Introduction

Heilongjiang Province is located in the black-soil zone of the Northeast Plain, one of the three major black-soil zones in the world. It is also the province with the largest rice planting area in China and occupies an important position in the world’s rice market [1,2]. The growth, development, and yield formation of rice are sensitive to temperature and light. Cold damage is one of the most important agrometeorological disasters for rice in cold regions. Rice cultivation in cold regions of China is mainly distributed in Heilongjiang Province [3,4], which belongs to both a high-cold rice region and mid–high latitude region, and the growing season of rice is susceptible to low temperature and cold damage [5,6,7,8]. According to the National Bureau of Statistics of China [9], from 2017 to 2019, the rice yield (kg/ha) in Heilongjiang Province was 7139.57, 7098.79, and 6986, respectively. Due to the impact of low temperature and cold damage, in recent years, rice yields in Heilongjiang Province have been declining at a rate of 1.59% [9]. The accumulated temperate zone is an important reference for the planting layout and variety selection of various crops in Heilongjiang Province [4]. Rice cultivation is mainly distributed in the 1st–3rd accumulated temperate zones of Heilongjiang Province. Different accumulated temperate zones have different requirements for GD of rice varieties. Choosing and planting rice varieties with suitable GD according to the accumulated temperate zone is an important measure to prevent low temperature and cold damage [10,11]. Japonica rice is a subspecies of rice that is resistant to low temperature and weak light, suitable for planting in Heilongjiang Province. It also has a low amylose content and good palatability [12]; therefore, rice breeding experts regard the accurate identification of GD of japonica rice varieties in cold regions as an important task.

Some researchers have obtained a variety of rice phenotype data through lots of field investigations, used statistical methods to analyze the phenotype data, and carried out research on the identification of rice GD [13,14,15,16]. The source of phenotypic data mostly relies on human experience, and the data collection process is time consuming and susceptible to environmental interference. Based on the identification results of phenotypic data, other researchers have carried out molecular QTL markers of rice variety GD to further confirm the molecules affecting rice GD [17,18,19,20]. Molecular labeling methods not only require skilled operation of instruments by professionals, but also incur high costs, which limits the development of large-scale molecular detection. With the developments in science and technology, there are more and more intelligent means. It is of great significance to design and construct an efficient, non-destructive, and accurate classification and identification method for japonica rice GD.

Raman spectroscopy technology can provide rapid, simple, repeatable, and non-destructive qualitative and quantitative analysis without sample preparation, which can be directly measured by laser [21]. Farber Charles [22] discussed the application of Raman spectra as an unlabeled, non-invasive, and non-destructive analysis technique for the rapid and accurate identification of nutrients in 15 different rice grains. Ling Zhu et al. [23] used Raman spectroscopy to identify the varieties and origin of rice in China. Tian [24] established a rapid non-destructive detection method for distinguishing rice-producing areas using Raman spectroscopy. Pezzotti Giuseppe’s team [25] analyzed polysaccharides from nine rice varieties in Japan based on Raman spectroscopy. At present, the application of Raman spectroscopy in the classification and identification of GD of japonica rice has not been reported.

Machine learning methods, such as principal component analysis (PCA), support vector machines (SVM) [26,27], random forests [28], and classification and regression trees (CART) [29], are often used in classification problems based on Raman spectral data [30,31,32,33,34,35]. However, they are limited by the shallow architecture of their own models, their performance is far inferior to machine deep learning methods, and the analysis of spectral data cannot be further improved [36]. In 2006, a paper by Hinton et al. [37] in *Science* led to the study of machine deep learning. Machine deep learning has been successfully applied to multiple pattern classification problems [38,39], including agricultural applications [40]. Convolutional neural networks (CNNs) [41,42] have been widely used as a classic kind of deep neural network. However, CNNs typically perform poorly with small datasets [43], which is the case for most of the plant database. In 2017, Hinton et al. [44] proposed a vector capsule network and a dynamic routing algorithm between capsules. Capsule networks establish the location relationship of features so that they can achieve better results than CNNs with similar structures in many small datasets [45]. In view of this, this paper was based on the structural advantages of CapsNets and its excellent performance. We adopted CapsNets architecture to solve the japonica rice classification problem.

At present, the application of Raman spectroscopy combined with CapsNets in the identification and classification of GD of japonica rice has not been reported. In this study, an intelligent classification model of japonica rice GD is proposed for the first time by combining Raman spectral data with CapsNets. The main purpose of this study was to provide an efficient and accurate intelligent classification method for japonica rice GD for breeding experts.

## 2. Materials and Methods

### 2.1. Test Material

Seven japonica rice varieties were collected from the experimental field of Qiqihar Branch of Heilongjiang Academy of Agricultural Sciences in September 2021. As shown in Figure 1, the seven tested japonica rice varieties were divided into three GD types (*p* < 0.05). QJ1 had a longer GD (144 days) and was suitable for planting in the first accumulation zone of Heilongjiang Province. SJ13 and HJ313 had a medium GD (134–135 days) and were suitable for planting in the second accumulation zone of Heilongjiang Province. LJ47, KY131, LJ11, and HH311 had short GD (124–126 days) and were suitable for planting in the third accumulation zone of Heilongjiang Province.

### 2.2. Measurement of Spectral Data

In September 2021, 10 holes of each variety were placed in a laboratory at room temperature of about 25 °C for natural air drying for 15 days, and the moisture content was reduced to about 20%. Five panicles were collected at different locations of each hole, and two grains were selected from each panicle, with a total of 100 grains from each variety. The husking process was carried out manually, and a thin blade was used to remove fluorescence on the surface of rice grains. Finally, 35 complete rice grains were selected as samples (Table 1), and a total of 245 samples were obtained.

The spectral data of samples were collected using an Advantage 532 Desktop Raman spectrometer. Each spectrum was scanned with a resolution of 1.4 cm^−1^ over 200–3400 cm^−1^, and the spectral information of 245 samples was obtained with 4 scanning times. ProScope HR 2.3 software was used to obtain sample image information and sample data, which were saved in PRN format. The data processing software used was Python.

### 2.3. Preprocessing of Spectral Data

Raman spectrum acquisition does not require sample preparation, but the sample shape, roughness, and acquisition parameters set in the acquisition software will also affect the results. In order to enhance data differences and promote the purpose of the modeling effect, spectrum preprocessing is essential. In this experiment, the original spectral data were filtered by the signal.filtfilt function [46]. The filter was constructed using signal.butter, where b, a= scipy.signal.butter (N, Wn) (N: the order of the filter; Wn: the critical frequency or frequencies; b: the numerator coefficient vector of the filter; a: the denominator coefficient vector of the filter) [47]. In this test, the N parameter was 2 (one-step forward and one-step backward filtering to avoid phase difference) and the Wn parameter was 0.002 (Wn = 2×cutoff frequency/sampling frequency). Secondly, based on the signal.filtfilt function, the difference method was used to extract the spectral crest information, as shown in Equation (1). Finally, the extracted spectral peak information was filtered by the signal.filtfilt function, the *N* parameter was set to 2, and the Wn parameter was set to 0.03.
(1)$$X=x-y_{1}$$

Note: $y_{1}$ is the spectral intensity filtered by the signal.filtfilt function, and x is the original spectral intensity.

### 2.4. Selection of Effective Crest Information

In order to better retain effective crest information, improve the model classification accuracy, and reduce the number of calculations, the selection of crest information is essential. Firstly, referring to wave crest extraction, Raman spectral characteristics, and the attribution of rice [48], seven effective crests were extracted at 480 cm^−1^, 865 cm^−1^, 941 cm^−1^, 1129 cm^−1^, 1339 cm^−1^, 1461 cm^−1^, and 2910 cm^−1^. Secondly, with each effective wave peak as the center, 56 and 55 points were taken forward and backward, respectively, and 112 points in total were taken as a window (Figure 2a). Each sample had 7 effective wave peaks, and a total of 784 points were extracted. Finally, the 112 points intercepted from each wave peak were divided into 4 segments with 28 points in length on average. Then, the four segments were aligned and overlapped (Figure 2b). Each wave peak obtained 28 × 4 two-dimensional data information, and each sample obtained 28 × 28 two-dimensional data information.

### 2.5. Evaluation Indices of CapsNets Model

The goal of this paper was to investigate and design a CapsNets model for classifying GD of japonica rice as accurately as possible. The realization of the CapsNets model is divided into three layers: initial layer (Conv1), primary capsules layer, and final CapsLayer (Figure 3).

As shown in Figure 3, Conv1 is a normal convolutional layer, which has 256 output channels and 9 × 9 convolution kernels with a stride of 1 and ReLU activation. This layer extracts low-level features that are then used as inputs to the primary capsules, a convolutional capsule layer. This convolutional layer has 256 output channels and 9 × 9 convolution kernels with a stride of 2. The outputs are segmented into [32 × 8] vectors (primary capsules). Therefore, each primary capsule output sees the outputs of all 256 × 9 × 9 Conv1 units, which is a group 8D vectors in the 6 × 6 grid. The primary capsules contain advanced features. The length of the output vector of a capsule represents the probability that the entity represented by the capsule is present in the current input. The output capsule is computed using a nonlinear squashing function (Equation (2)) to ensure that the length of the vector is between 0 and 1. The third layer (final CapsLayer) has one capsule per class, and each of these capsules receives input from all the capsules in the layer below.
(2)$$v_{j}=\frac{{\left\|s_{j}\right\|}^{2}}{1+{\left\|s_{j}\right\|}^{2}}\frac{s_{j}}{\left\|s_{j}\right\|}$$

Note: $v_{j}$ is the vector output of capsule *j*, and $s_{j}$ (Equation (3)) is the input vector of capsule *j*.
(3)$$s_{j}=\sum_{i}c_{ij}{\hat{u}}_{\left(j|i\right)}$$

Note: $c_{ij}$ (Equation (6)) are coupling coefficients. ${\hat{u}}_{\left(j|i\right)}$ (Equation (5)) is the prediction vector of the output of capsule *j* at a higher level computed by capsule *i* in the layer below.

Agreement routing is used between primary capsules and final CapsLayer. Agreement $a_{ij}$ Equation (4) for updating log probabilities and coupling coefficients is calculated (Equation (6)).
(4)$$a_{ij}=v_{j}{\hat{u}}_{\left(j|i\right)}$$

Note: $v_{j}$ is the vector output of capsule *j*. ${\hat{u}}_{\left(j|i\right)}$ is calculated.
(5)$${\hat{u}}_{\left(j|i\right)}=W_{ij}u_{i}$$

Note: $u_{i}$ is the output of capsule *i* in the layer below. *W*_*ij*_ is a weight matrix between each $u_{i}$ in primary capsules and $v_{j}$, *j* ∈ (1, 3), which needs to be learned in the back pass. Coupling coefficients are calculated as follows:(6)$$c_{ij}=\frac{\exp(b_{ij})}{\sum_{k}\exp(b_{ij})}$$

Note: $c_{ij}$ are coupling coefficients. $b_{ij}$ is the log probability that capsule *i* will be coupled with capsule *j* and it is initially set to 0 at the beginning of the routing by agreement process. $c_{ij}$ is calculated from $b_{ij}$ using the softmax function.

The final CapsLayer work can be simply summarized into four steps: I, matrix transformation; II, input weighting (Equation (5)); III, weighted sum (Equations (3)–(6)); IV, nonlinear transformation (Equation (2)). We use the Python-based pytorch (a deep learning algorithm framework) to build the CapsNets to complete the experiment.

## 3. Results and Discussion

### 3.1. Analysis of Spectral Data Preprocessing

Figure 4 shows that the spectral information of the seven rice varieties (245 samples) was intertwined in a disorderly manner, making it difficult to distinguish. Therefore, it was essential to preprocess the spectral data.

Figure 5 shows that the filtered curve by the signal.filtfilt function was smoother and clearer than the original curve. The normalization algorithm was used to eliminate the dimensional influence [49] and optimize the spectral data [49], and the Wn (critical frequency or frequencies) parameter of this experiment was set to 0.002. This parameter was able to filter more details and invalid information as much as possible, and reserve more original information for the difference method to extract wave peaks.

As shown in Figure 6, this experiment used the difference method to obtain 10 obvious peaks, which were extracted at 480 cm^−1^, 865 cm^−1^, 941 cm^−1^, 1129 cm^−1^, 1339 cm^−1^, 1461 cm^−1^, 1780 cm^−1^, 2140 cm^−1^, 2330 cm^−1^, and 2910 cm^−1^. At the same time as the peaks were obtained, small spectral clutters appeared. Therefore, the signal.filtfilt function was used to filter the peaks again.

Figure 7 shows the curve of the peak information processed by the difference method. The scipy.signal.filtfilt method was smoother and clearer than that processed by the difference method. When filtering wave peaks, the Wn parameter 0.03 was selected, which filtered out invalid information again and retained more original effective information, thus improving the resolution of the identification model.

### 3.2. Analysis of Selection of Effective Crest Information

In this experiment, two segmentation methods and one superposition fusion method were used to convert one-dimensional data to two-dimensional data, and 245 2D data of 28 × 28 were obtained. The imshow method [50] in the Python-based matplotlib package was used to complete the data visualization (Figure 8). Two-dimensional scalar data are presented as a pseudocolor image. The values were mapped to colors using normalization and a color map. Based on the fusion of local features and global features [51], the two-dimensional scalar data were transformed into two-dimensional visual image information, and the recognition rate of japonica rice GD identification was improved by taking advantage of CapsNets’ ability in image information classification [45].

### 3.3. Performance Analysis of CapsNets Model

In order to study the modeling performance of the CapsNets model for japonica rice GD identification and classification, Python was first used to eliminate abnormal sample image information (Table 2), and then 177 labeled items of image sample information of the training set were brought into the CapsNets model for machine deep learning. Finally, 56 items of image sample information of the test set were brought into the CapsNets model for identification, and a CapsNets total of 160 training sessions were conducted. The results are shown in Figure 9.

As shown in Figure 9, there was no significant change in the accuracy of the training datasets and test datasets when the training epochs were between 10 and 35. The accuracy of the training datasets and test datasets improved significantly when the training epochs were between 35 and 110. The accuracy fluctuated significantly when the training epochs were between 110 and 120. Finally, the accuracy converged to 86–92.85% when the training epochs were between 120 and 160.

Table 3 shows the training epochs, value of loss function, and accuracy of training datasets when the accuracy of test datasets exceeded 90%. When the training epochs were 133 and 150, the accuracy of test datasets reached the highest of 93%, and when the training epochs were 160, the accuracy of training datasets reached the highest of 89%. When the values of loss function were 0.1074 and 0.1181, the accuracy of test datasets was up to 93%, and when the value of loss function was 0.1014, the accuracy of training datasets was up to 89%.

### 3.4. Discussion of the Description Results

The GD of rice is related to the amylum content in grains [52,53]. Referring to the results of Raman spectral correlation analysis, it can be seen that the effective band range of rice mainly concentrates in the region of 200–1800 cm^−1^ and 2800–3200 cm^−1^ [54,55]. In this experiment, wave crests were extracted from Raman spectral data of samples at GD stage by signal.filtfilt-difference method. In combination with the Raman spectral correlation analysis of rice, seven wave crests were extracted, and the attributed substances were amylum and sugar [48,56], with amylum being the aggregation of sugar molecules [57,58]. The spectral characteristic frequency of the same species with different attributes had different offsets, i.e., the spectral crest image had different offsets [59]. In this experiment, 1/2 of each wave crest was selected for segmentation, in order to extract the characteristic information of the wave crest offset of each sample. The same species owned a wave crest at the same Raman shift, but the wave crest starts, ends, and peaks of samples with different attributes were different [48,59]. In this experiment, 1/4 and 3/4 of each wave crest were selected for segmentation processing, in order to extract the characteristic information of the start, end, and peak intensity of each sample. In view of this, the seven wave crests extracted from each sample and the characteristic information selected by the segmentation method for each wave crest in this experiment provided a basis for the classification of biological characteristics of rice GD.

The capsule network suitable for small datasets is improved by modifying the dynamic routing protocol. Agreement routing imitates hierarchical communication of information across neurons in human brains that are responsible for perception and understanding [43]. For each potential parent, the capsule network can increase or decrease the connection strength by dynamic routing, which is more effective than primitive routing strategies, such as max pooling in CNNs [60]. Considering the number of samples (233) in this experiment, we used a full-link routing protocol. In other words, the output was routed to all possible parents. Table 3 shows that in the small sample set, when the training cycles were 110–160 and the loss function was 0.1014–0.1239, the training set and test set had the highest performance: 89% and 93%, respectively. In the few-shot task (Figure 10), the samples are regarded as parts, the classes are regarded as the whole, and the class representation encoded by the dynamic routing algorithm is more representative [61].

Overfitting is already an unavoidable problem in machine deep learning on small sample datasets, which can be solved from three aspects: training data, training process, and model structure [62]. Dataset enhancement based on image transformation can effectively improve the overfitting problem of image classification models [63], but the two-dimensional data in this experiment were generated through data transformation, so this method was not applicable. The generation of the overfitting problem is closely related to the training process. As shown in Figure 11, during the training process, the model experienced underfitting (epoch less than 120) to overfitting (epoch greater than 170). In this experiment, the method of early stopping was adopted, and selecting the appropriate number of training cycles (the epoch was equal to 160) effectively avoided the overfitting problem. It is also possible to use the L1 (lasso) regularization or L2 (ridge) regularization method to process the weight parameters during the training process to solve the problem of overfitting [64,65]. Because the capsule network model in this experiment used the capsule structure and dynamic routing, this method was not used. An overly complicated model structure is also a major cause of overfitting. Therefore, simplifying the model structure, reducing the network capacity, and reducing the number of parameters can also effectively prevent overfitting. In this experiment, the structure of the three-layer model was used (Figure 11). Even though the number of training cycles continued to increase, the difference between the accuracy of the model on the training set and the test set did not continue to expand, but remained within a reasonable interval.

## 4. Conclusions

The present study exhibits the feasibility of Raman spectroscopy combined with the CapsNets method for the classification of japonica rice GD. In order to improve the classification accuracy, a method of preprocessing of spectral data was constructed based on signal.filtfilt combined with difference and signal.filtfilt. For the selection of effective crests, information was obtained based on two segmentation methods and one superposition fusion method, and the CapsNets model was established. The training epochs were between 110 and 160, and the classification accuracy of the CapsNets model was between 91% and 93%. The performance of the CapsNets model tended to be stable. It provides a new approach to establish an efficient and accurate intelligent classification method of japonica rice GD.

## Figures and Tables

**Figure 1 plants-11-01573-f001:**
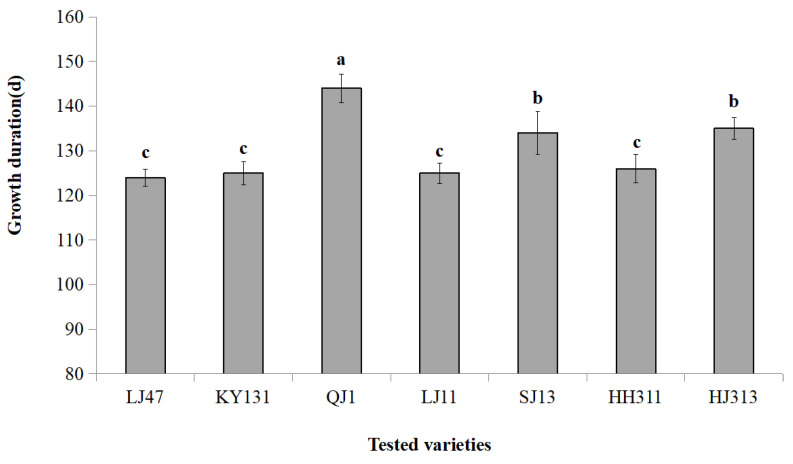
The GD values of 7 tested japonica rice varieties. Values with different superscript letters were significantly different at *p* < 0.05. The GD refers to the number of days (d) that rice takes from emergence to maturity.

**Figure 2 plants-11-01573-f002:**
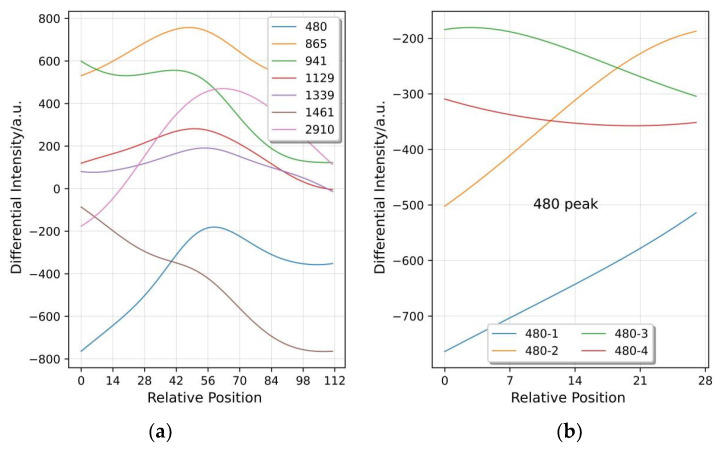
(**a**) Seven wave peak curves were extracted from each sample. (**b**) Each wave peak is divided into four sections of the curve.

**Figure 3 plants-11-01573-f003:**
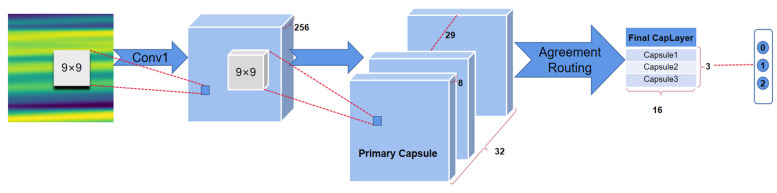
Schematic of the CapsLayer model.

**Figure 4 plants-11-01573-f004:**
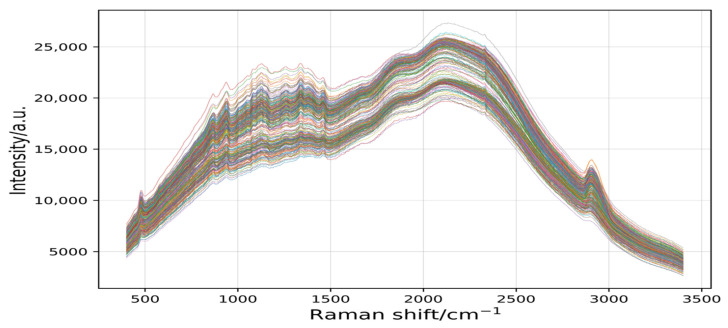
The 245 raw spectral curves for 7 varieties. Raman shift is the reciprocal of wavelength, and its range is 200–3400 cm^−1^. Intensity is the intensity of Raman scattering.

**Figure 5 plants-11-01573-f005:**
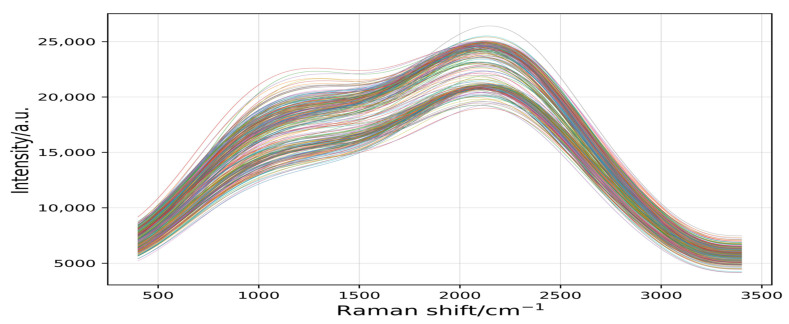
The 245 spectral curves were filtered by the signal.filtfilt function.

**Figure 6 plants-11-01573-f006:**
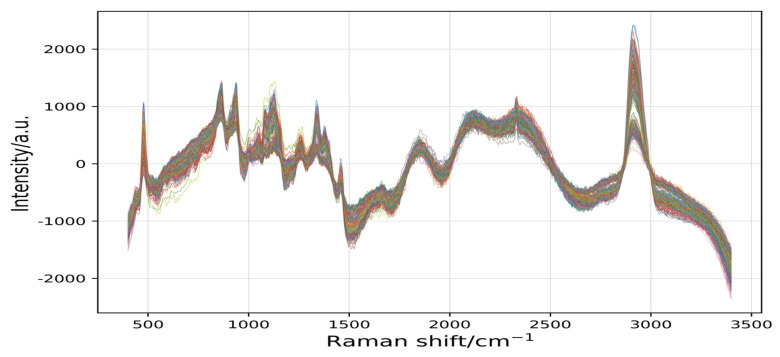
The curve of the peak information extracted by the difference method for 245 samples.

**Figure 7 plants-11-01573-f007:**
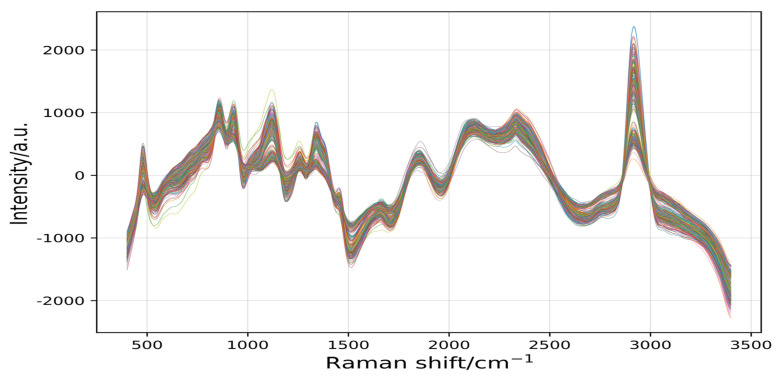
The curves of the peaks were filtered by the signal.filtfilt function for 245 samples.

**Figure 8 plants-11-01573-f008:**
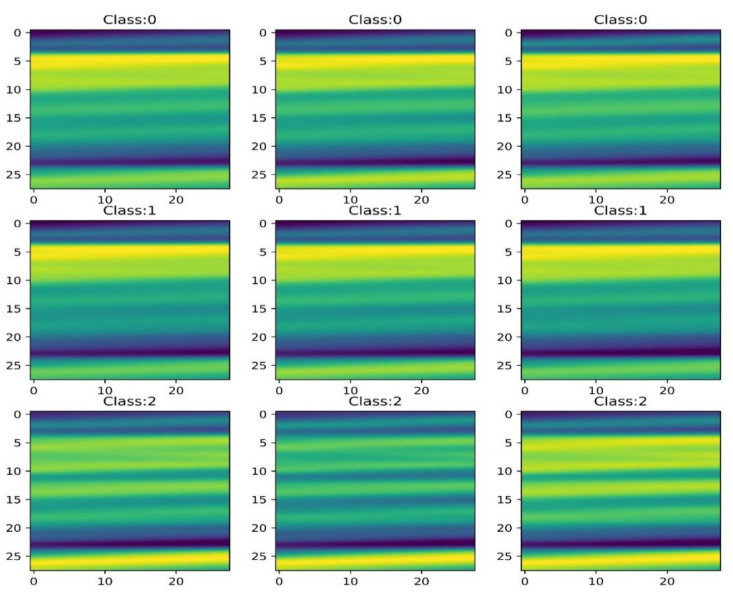
Nine pieces of 28 × 28 two-dimensional image information. Class 0, Class 1, and Class 2 were japonica rice varieties suitable for planting in the third, first, and second accumulation zones of Heilongjiang Province, respectively.

**Figure 9 plants-11-01573-f009:**
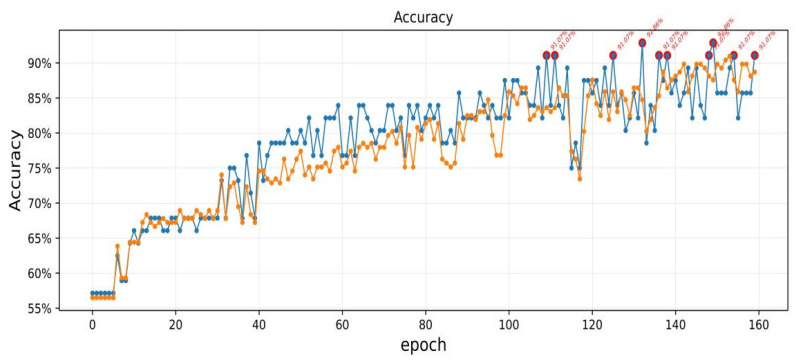
Classification results of japonica rice GD by CapsNets model. Epoch represents the number of training cycles of the model, and the yellow line and blue line represent the accuracy of training datasets and test datasets, respectively.

**Figure 10 plants-11-01573-f010:**
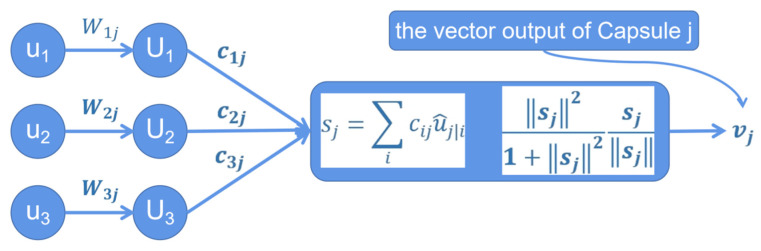
Schematic diagram of neural capsule calculation process.

**Figure 11 plants-11-01573-f011:**
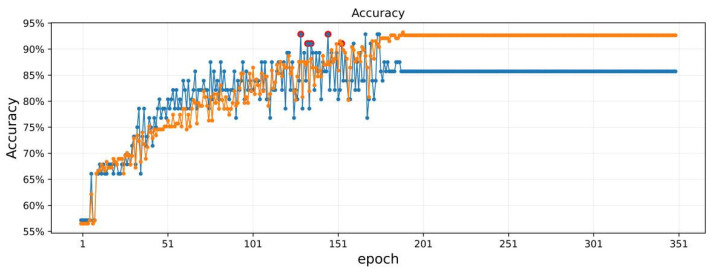
Classification results of japonica rice GD by CapsNets model. Epoch was the number of training cycles of the model, and the yellow line and blue line represent the accuracy of the training and test datasets, respectively.

**Table 1 plants-11-01573-t001:** Appropriate accumulated temperature zone and quantity of test samples (ATZH: accumulation temperate zone in Heilongjiang Province).

Serial Number	Name of Sample	ATZH	Number of Samples
1	QJ1	The first	35
2	SJ13	The second	35
3	HJ313	The second	35
4	LJ47	The third	35
5	KY131	The third	35
6	LJ11	The third	35
7	HH311	The third	35

**Table 2 plants-11-01573-t002:** Number of training samples and test samples with classification labels.

Name of Sample	Label of Sample	Sample		Subtotal
		Training Set	Test Set	
LJ47	0	24	8	32
KY131	0	26	8	34
LJ11	0	24	8	32
HH311	0	26	8	34
QJ1	1	26	8	34
SJ13	2	26	8	34
HJ313	2	25	8	33
Total		177	56	233

**Table 3 plants-11-01573-t003:** The training cycles and loss function values with high accuracy were obtained by the CapsNets model.

Serial Number	Training Epochs	Value of Loss Function	Accuracy of Training Datasets	Accuracy of Test Datasets
1	110	0.1148	84	91
2	112	0.1239	84	91
3	126	0.1111	86	91
4	133	0.1181	86	93
5	137	0.1194	85	91
6	139	0.1157	86	91
7	149	0.1076	88	91
8	150	0.1074	88	93
9	155	0.1101	88	91
10	160	0.1014	89	91

## Data Availability

The data and code presented in this study are openly available at: https://github.com/zxxsnh/Rice_GD_classification_CapsNets (accessed on 14 June 2022).

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
