# Peer review of "Intelligent Classification of Japonica Rice Growth Duration (GD) Based on CapsNets"

_plants, 2022, doi:10.3390/plants11121573_

Round 1
Reviewer 1 Report
Specific comments are provided as follows
1. Spell out “GD” when it appears at the first time in the manuscript.
2. Line 136 “N parameter was set to 2, Wn parameter was set to 0.03.” Why to set these numbers? What number was set for a: the denominator coefficient vector of the filter and b: the numerator coefficient vector of the filter?
3. Why to choose capsule networks? If the reason is the context that the authors addressed on line 262-277, this paragraph should be moved to the introduction section to explain the reason choosing capsule networks rather than other deep learning models. Also, this paragraph reviews Convolutional neural network (CNNs) applied to agriculture classification. “Deep learning techniques to classify agricultural crops through UAV imagery: a review” and “Semantic Segmentation Using Deep Learning with Vegetation Indices for Rice Lodging Identification in Multi-date UAV Visible Images” could be proper references related to the application of deep neural network to classify rice patterns.
4. In “3.4. Discussion of the Description Results” more results derived from this study should be discussed besides a general description about CNNs, such as model modification and quantitative expression about its advantage, rather than literature review. How well Capsule Networks performed in this study?
5. An over-fitting problem during model training often occurs on small datasets. How to overcome this problem, in addition to judging the accuracy difference between the training datasets and the test datasets at the end of each epoch in the training process to control the training process?
6. “4. Conclusions” should be enhanced by adding more concrete results derived from this study.
Author Response
Dear Reviewer:
Thank you for your letter and for the reviewers’ comments concerning our manuscript entitled “plants-1738626”. Those comments are all valuable and very helpful for revising and improving our paper, as well as the important guiding significance to our researches. We have studied comments carefully and have made correction which we hope meet with approval. Especially the suggestions in the discussion and conclusion sections are of great help to improve the quality of the paper. Revised portion are marked in the yellow highlight in the paper. The main corrections in the paper and the responds to the reviewer’s comments are as following:
Point 1: Spell out “GD” when it appears at the first time in the manuscript.
Response 1: Title “Intelligent Classification of Japonica Rice Growth Duration (GD) Based on CapsNets”.
Point 2: Line 136 “N parameter was set to 2, Wn parameter was set to 0.03.” Why to set these numbers? What number was set for a: the denominator coefficient vector of the filter and b: the numerator coefficient vector of the filter?
Response 2: According to the suggestions of reviewer, we have made the following supplements to signal.butter of signal.filtfilt function. “In this test, N parameter was 2 (one-step forward and one-step backward filtering to avoid phase difference) and Wn parameter was 0.002 (Wn=2*cut-off frequency/sampling frequency).”
Point 3: Why to choose capsule networks? If the reason is the context that the authors addressed on line 262-277, this paragraph should be moved to the introduction section to explain the reason choosing capsule networks rather than other deep learning models. Also, this paragraph reviews Convolutional neural network (CNNs) applied to agriculture classification. “Deep learning techniques to classify agricultural crops through UAV imagery: a review” and “Semantic Segmentation Using Deep Learning with Vegetation Indices for Rice Lodging Identification in Multi-date UAV Visible Images” could be proper references related to the application of deep neural network to classify rice patterns.
Response 3: According to the suggestions of reviewer, we have added more supplement. Such as the introduction: “ Machine learning methods such as principal components analysis (PCA), support vector machines (SVM) [27,28], random forests [29], classification and regression tree (CART) [30] are often used in classification problems based on Raman spectral data [31-35]. However, they are limited by the shallow architecture of their own models, their performance is far inferior to machine deep learning methods, and the analysis of spectral data cannot be further improved [37]. Since 2006 Hinton et al. [38] posted in Science led to the study of machine deep learning. Machine deep learning has been successfully applied to multiple pattern classification problems [39,40] including agricultural applications [41]. Convolutional neural network (CNNs) [42,43] that has been widely used is a classic kind of deep neural network. But CNNs typically perform poorly confronting small datasets [44], which is the case for most of the plant database. In 2017, Hinton et al. [45] proposed a vector capsule network and a dynamic routing algorithm between capsules. Capsule networks established the location relationship of features so that it can achieve better results than CNNs with similar structure in many small datasets [46]. In view of this, this paper was based on the structural advantages of CapsNets and its excellent performance, we adopt the CapsNets architecture for maturity of japonica rice classification problem.” References 41 and 43 were cited.
Point 4: In “3.4. Discussion of the Description Results” more results derived from this study should be discussed besides a general description about CNNs, such as model modification and quantitative expression about its advantage, rather than literature review. How well Capsule Networks performed in this study?
Response 4: According to the suggestions of reviewer, we have added more information about the second paragraph of” 3.4. Discussion of the Description Results.” “The capsule network suitable for small data sets is improved by modifying the dynamic routing protocol. The agreement-routing imitates hierarchical communication of information across neurons in human brains that are responsible for perception and understanding [44]. For each potential parent, the capsule network can increase or decrease the connection strength by dynamic routing, which is more effective than the primitive routing strategies such as max-pooling in CNNs [62]. Considering the number of samples (233) in this experiment, this paper used a full-link routing protocol, in other words the output is routed to all possible parents. Table 3 showed that in the small sample set, when the training times were between 110-160 and the loss function was between 0.1014-0.1239, the training set and test set had the highest performance, 89% and 93%, respectively. (Figure 10) in the few-shot task, the samples are regarded as parts and the classes are regarded as the whole, and the class representation encoded by the Dynamic Routing algorithm is more representative [63]”
Point 5: An over-fitting problem during model training often occurs on small datasets. How to overcome this problem, in addition to judging the accuracy difference between the training datasets and the test datasets at the end of each epoch in the training process to control the training process?
Response 5: Thank you very much for the suggestions of the reviewers on the Discussion. We rewrote this section and discussed a large body of literature. As discussed in the third part” Overfitting is an unavoidable problem in machine deep learning on small sample data sets, the overfitting problem could be solved from three aspects: training data, training process and model structure [64]. Data set enhancement based on image transformation can effectively improve the overfitting problem of image classification models [65], but the two-dimensional data in this experiment was generated through data transformation, so this method was not applicable. The generation of the overfitting problem is closely related to the training process. As shown in Figure 11, during the training process, the model has experienced from underfitting (epoch less than 120) to overfitting (epoch greater than 170). In this experiment, the method of early stopping was adopted, and selecting the appropriate number of training times (the epoch was equal to 160) could effectively avoid the overfitting problem. It is also possible to use the L1 (lasso) regularization or L2 (ridge) regularization method to process the weight parameters during the training process to solve the problem of overfitting [66,67], because the capsule network model in this experiment uses the capsule structure and dynamic routing, so this method was not used. The over-complicated model structure is also a major cause of over-fitting. Therefore, simplifying the model structure, reducing the network capacity, and reducing the number of parameters can also effectively prevent over-fitting. In this experiment, the structure of the three-layer model was used, (Figure 11) even if the number of trainings continues to increase, the difference between the accuracy of the model on the training set and the test set did not continue to expand, but remained within a reasonable interval. ” and Figure 11.
Point 6: “4. Conclusions” should be enhanced by adding more concrete results derived from this study.
Response 6: Thank you very much for the suggestions of the reviewers on the Conclusions. Increased”The present study exhibits the feasibility of Raman spectroscopy combine with CapsNets method for classification of japonica rice GD. In order to improve the classification accuracy, a method of preprocessing of spectral data was constructed based on signal.filtfilt combined with difference and signal.filtfilt. Selection of effective crest Information was carried out based on two segmentation methods and one superposition fusion method, and the CapsNets model was established. The Training epochs was between 110 and 160, and the classification accuracy of CapsNets model was between 91% and 93%. The performance of the CapsNets model tended to be stable. It provides a new approach to establish an efficient and accurate intelligent classification method of japonica rice GD.”
Once again, thank you very much for your good comments and suggestions.

Reviewer 2 Report
The paper addresses an interesting area of study. However, there are issues which currently raise some concerns regarding the presented study:
- In my opinion, the literature review is presented to show existing works and to bridge the gap in knowledge. Authors have shown few existing works and have not addressed the gap in knowledge either. The authors should separate the introduction from the literature review.
- The authors should better explain de existing particularities of japonica rice in order to motivate the goal of this research.
- The authors have presented details on the used neural network but, according to the presented results, the net is overfitted. In this case, the accuracy values ​​are flawed. The authors should better discuss how the number of epochs, defined as a hyper-parameter, influence the performance of the deep learning model. Also, the validation loss is not presented. It is well known that the variation of the number of epochs stops when the validation loss does not longer improve
- The paper fails to provide a comparison with the reported results in the literature.
Minor remarks
Please clarify what 6*6 grid means in this work. Fig. 3 mentions another 9x9 kernel. Is the notation x similar to *? In the case the answer is ‘yes’ then the Figure 3 and the related comments should be amended.
The quality of English language should be improved.
Author Response
Dear Reviewer:
Thank you for your letter and for the reviewers’ comments concerning our manuscript entitled “plants-1738626”. Those comments are all valuable and very helpful for revising and improving our paper, as well as the important guiding significance to our researches. We have studied comments carefully and have made correction which we hope meet with approval. Revised portion are marked in the yellow highlight in the paper. The main corrections in the paper and the responds to the reviewer’s comments are as following:
Point 1: In my opinion, the literature review is presented to show existing works and to bridge the gap in knowledge. Authors have shown few existing works and have not addressed the gap in knowledge either. The authors should separate the introduction from the literature review.
Response 1: According to the suggestions of reviewers, we separate the introduction from the literature review. We rewrite " Machine learning methods such as principal components analysis (PCA), support vector machines (SVM) [27,28], random forests [29], classification and regression tree (CART) [30] are often used in classification problems based on Raman spectral data [31-35]. However, they are limited by the shallow architecture of their own models, their performance is far inferior to machine deep learning methods, and the analysis of spectral data cannot be further improved [37]. Since 2006 Hinton et al. [38] posted in Science led to the study of machine deep learning. Machine deep learning has been successfully applied to multiple pattern classification problems [39,40] including agricultural applications [41]. Convolutional neural network (CNNs) [42,43] that has been widely used is a classic kind of deep neural network. But CNNs typically perform poorly confronting small datasets [44], which is the case for most of the plant database. In 2017, Hinton et al. [45] proposed a vector capsule network and a dynamic routing algorithm between capsules. Capsule networks established the location relationship of features so that it can achieve better results than CNNs with similar structure in many small datasets [46]. In view of this, this paper was based on the structural advantages of CapsNets and its excellent performance, we adopt the CapsNets architecture for maturity of japonica rice classification problem." in the fourth part of the preface and " The capsule network suitable for small data sets is improved by modifying the dynamic routing protocol. The agreement-routing imitates hierarchical communication of information across neurons in human brains that are responsible for perception and understanding [44]. For each potential parent, the capsule network can increase or decrease the connection strength by dynamic routing, which is more effective than the primitive routing strategies such as max-pooling in CNNs [62]. Considering the number of samples (233) in this experiment, this paper used a full-link routing protocol, in other words the output is routed to all possible parents. Table 3 showed that in the small sample set, when the training times were between 110-160 and the loss function was between 0.1014-0.1239, the training set and test set had the highest performance, 89% and 93%, respectively. (Figure 10) in the few-shot task, the samples are regarded as parts and the classes are regarded as the whole, and the class representation encoded by the Dynamic Routing algorithm is more representative [63]" in 3.4 discussion part.
Point 2: The authors should better explain de existing particularities of japonica rice in order to motivate the goal of this research.
Response 2: As suggested by reviewers, we added japonica Rice's description in the first paragraph of Introduction. As follows.” Japonica rice is a subspecies of rice, it is resistant to low temperature and weak light, suitable for planting in Heilongjiang Province, and japonica rice has low amylose content and good palatability [13],”
Point 3: The authors have presented details on the used neural network but, according to the presented results, the net is overfitted. In this case, the accuracy values ​​are flawed. The authors should better discuss how the number of epochs, defined as a hyper-parameter, influence the performance of the deep learning model. Also, the validation loss is not presented. It is well known that the variation of the number of epochs stops when the validation loss does not longer improve.
Response 3: According to the suggestions of reviewer, we have added more information about the third paragraph of 3.4. “Overfitting is an unavoidable problem in machine deep learning on small sample data sets, the overfitting problem could be solved from three aspects: training data, training process and model structure [64]. Data set enhancement based on image transformation can effectively improve the overfitting problem of image classification models [65], but the two-dimensional data in this experiment was generated through data transformation, so this method was not applicable. The generation of the overfitting problem is closely related to the training process. As shown in Figure 11, during the training process, the model has experienced from underfitting (epoch less than 120) to overfitting (epoch greater than 170). In this experiment, the method of early stopping was adopted, and selecting the appropriate number of training times (the epoch was equal to 160) could effectively avoid the overfitting problem. It is also possible to use the L1 (lasso) regularization or L2 (ridge) regularization method to process the weight parameters during the training process to solve the problem of overfitting [66,67], because the capsule network model in this experiment uses the capsule structure and dynamic routing, so this method was not used. The over-complicated model structure is also a major cause of over-fitting. Therefore, simplifying the model structure, reducing the network capacity, and reducing the number of parameters can also effectively prevent over-fitting. In this experiment, the structure of the three-layer model was used, (Figure 11) even if the number of trainings continues to increase, the difference between the accuracy of the model on the training set and the test set did not continue to expand, but remained within a reasonable interval.”
Figure 11. Classification results of japonica rice GD by CapsNets model. Epoch was the number of training times of the model, and the yellow line and blue line represented the accuracy of training datasets and test datasets respectively.
Point 4: The paper fails to provide a comparison with the reported results in the literature.
Response 4: Thank you very much for the suggestions of the reviewers on the Conclusions. Increased”The present study exhibits the feasibility of Raman spectroscopy combine with CapsNets method for classification of japonica rice GD. In order to improve the classification accuracy, a method of preprocessing of spectral data was constructed based on signal.filtfilt combined with difference and signal.filtfilt. Selection of effective crest Information was carried out based on two segmentation methods and one superposition fusion method, and the CapsNets model was established. The Training epochs was between 110 and 160, and the classification accuracy of CapsNets model was between 91% and 93%. The performance of the CapsNets model tended to be stable. It provides a new approach to establish an efficient and accurate intelligent classification method of japonica rice GD. ”
Point 5: Please clarify what 6*6 grid means in this work. Fig. 3 mentions another 9x9 kernel. Is the notation x similar to *? In the case the answer is ‘yes’ then the Figure 3 and the related comments should be amended.
Response 5: In view of this problem, we modified as follows. ”As shown in Figure 3, Conv1 is a normal convolutional layer, which has 256 output channels, 9*9 convolution kernels with a stride of 1 and ReLU activation. This layer extracting low level features that are then used as inputs to the Primary Capsules. The Primary Capsules is a convolutional capsule layer. This convolutional layer has 256 output channels, 9*9 convolution kernels with a stride of 2. The outputs are segmented into [32×8] vectors (Primary Capsules). So, each Primary Capsules output sees the outputs of all 256*9*9 Conv1 units, which is a group 8D vectors in the 6*6 grid.” and Figure 3.
Figure 3. The schematic of the CapsLayer Model.
Point 6: The quality of English language should be improved.
Response 6: Thanks for the reviewer's suggestions on English language polish, we are in contact with the editor.
Once again, thank you very much for your good comments and suggestions.

Round 2
Reviewer 1 Report
1. In "Figure 9. Classification results of japonica rice GD by CapsNets model", the figures of top accuracy can be clearly marked to avoid overlapping.
2. Line 313-315 grammatic error.
3. Line 344 "The Training epochs was between 110 and 160" . should be "were".
4. Reference format should be checked for consistency.
Author Response
Dear Reviewer:
Thank you again for your letter and the reviewers for their comments on our manuscript entitled "plants-1738626". These comments are very helpful for the further revision and improvement of our paper, and have important guiding significance for our research. The main corrections of the paper and the responses to the reviewers' comments are as follows:
Point 1: In "Figure 9. Classification results of japonica rice GD by CapsNets model", the figures of top accuracy can be clearly marked to avoid overlapping.
Response 1: In response to this problem, we used Python to redraw the drawing, as shown in Figure 9.
Point 2: Line 313-315 grammatic error.
Response 2: Based on the questions raised by the reviewers, we mad grammatical corrections to this sentence, as follown.” Overfitting is already an unavoidable problem in machine deep learning on small sample data sets, which can be solved from three aspects: training data, training process and model structure [64].”
Point 3: Line 344 "The Training epochs was between 110 and 160" . should be "were".
Response 3: Based on the questions raised by the reviewers, we have made corrections, such as “The Training epochs were between 110 and 160, and the classification accuracy of CapsNets model was between 91% and 93%.”
Point 4: Reference format should be checked for consistency.
Response 4: Thank you very much for the Reference format, we revised seven issues in the reference, such as:
- Song, L.J. Study on nitrogen nutrition diagnosis for cold-terra rice based on airborne multispectral imager and SPAD [D]. Shenyang Agricultural University: Shenyang, China, DOI: 10.27327/d.cnki.gshnu.2020.000011.
- Wang, X.P. Effects of Drought Stress at Booting Stage on Carbon Metabolism and Yield Formation of Japonica Rice in Cold Region [D]. Northeast Agricultural University: Harbin, China, 2020. DOI: 10.27010/d.cnki.gdbnu.2020.000020.
- Feng, Y.J. Effect of water and nitrogen coupling on growth and nutrient absorption of japonica rice in cold region [D]. Shenyang Agricultural University: Shenyang, China,2017.
- Random forests[J]. MACH LEARN, 2001, 2001,45(1) (-):5-32. DOI:10.1023/A:1010933404324.
- Zhang, H.; Ma, B.J.; Zhang, C.M.; Zhao, B.H.; Xu, J.; Shao, S.M.; Gu, J.F.; Liu, L.J.; Wang, Z.Q.; Yang,J.C.Effects of alternate wetting and drying irrigation during whole growing season on quality and starch properties of rice[J].Journal of Yangzhou Univesity (Agricultural and Life Science Edition): Jiangsu, China,2020,41(06):1-8.DOI:10.16872/j.cnki.1671-4652.2020.06.001.
- Tian, F.M. Identification of Rice Based on Analysis of Raman spectrum and Organic Ingredients[D]. Jilin University: Jilin, China, DOI: 10.3390/PLANTS10061088
- Tan, F. Research on the Spectral Characteristics and Early Detection Mechanism of Rice Blast in Cold Area[D].Heilongjiang Bayi Agricultural University: Daqing, China,
There are some other minor issues, which have been revised in the article. Again, thank you very much for your good comments and suggestions.

Reviewer 2 Report
Authors have incorporated all the suggestions made. The paper is in much better shape now and may be accepted for publication.
Author Response
Dear Reviewer:
There are some other minor issues, which have been revised in the article. Again, thank you very much for your good comments and suggestions.